# Knowledge gaps and positive attitudes toward adult vaccination among nursing students: A Cross-sectional study

Noelia Rodríguez-Blanco[1], Nancy Vicente-Alcalde[2], Cristina Orts-Ruiz[3], Sergio Montero-Navarro[3], Cristina Salar-Andreu[3], Jesús Sánchez-Más[4]*, José Luis Duro-Torrijos[1,5]

1 Department of Nursing, Research Group Quality of Life and Health, Faculty of Health Sciences, Universidad Europea de Valencia SLU, Elche, Spain, 2 Nursing Department, Health Sciences Faculty, CEU-Cardenal Herrera University, CEU Universities, Elche, Spain, 3 Physical Therapy Department, Health Sciences Faculty, CEU-Cardenal Herrera University, CEU Universities, Elche, Spain, 4 Biomedical Sciences Department, Health Sciences Faculty, CEU-Cardenal Herrera University, CEU Universities, Plaza Reyes Católicos, Elche, Spain, 5 Department Research and Teaching, Vinalopo University Hospital, Elche, Spain

* jesus.sanchez@uchceu.es

## Abstract

### Background

Vaccination is one of the most effective public-health measures, yet adult coverage in Spain remains suboptimal due to misinformation and hesitancy. Nursing students, as future immunization promoters, play a pivotal role in vaccine recommendations. This study explored knowledge, attitudes, and recommendation practices regarding adult vaccination among undergraduate nursing students.

### Methods

An observational, descriptive, cross-sectional study was conducted in the province of Alicante, Spain, between April and May 2025. An ad hoc electronic survey with 19 closed-ended items was distributed to students from the three universities offering the Nursing degree in this province; 562 students participated. Differences across academic years were analyzed using Pearson's $\chi^2$ test for categorical variables and the Kruskal-Wallis test for the continuous variable age (non-normally distributed).

### Results

Overall, 39.1% of nursing students were unaware of the Spanish adult vaccination schedule with significant differences across academic years (55.7% in first-year vs. 27.2% in fourth-year students; $p < 0.001$). Only 44.1% reported sufficient training on vaccines. Attitudes toward adult vaccination were predominantly positive: 61.4% highly favorable, 62.5% highly effective, 66.7% supporting mandatory vaccination.

**Data availability statement:** All relevant data are within the manuscript and its Supporting Information files.

**Funding:** The author(s) received no specific funding for this work.

**Competing interests:** The authors have declared that no competing interests exist.

Additionally, 65.5% would recommend all scheduled adult vaccines. However, 47.9% perceived resistance from other healthcare professionals when recommending vaccines (p<0.001 for the increase across academic years).

## Conclusions

Nursing students in Alicante exhibit positive attitudes toward adult vaccination, yet significant knowledge gaps regarding the Spanish adult vaccination schedule persist even in the final year. Targeted curricular reinforcement from the first academic year is warranted within these three institutions to strengthen their role as immunization promoters.

## 1. Introduction

The World Health Organization (WHO) estimates that vaccination prevents approximately 3.5 to 5 million deaths annually from vaccine-preventable diseases worldwide [1]. However, despite the recognized success of vaccines, adult vaccination coverage remains lower than that in children [2,3]. It is evident that more than half of preventable diseases in the adult population, and potentially up to 90%, could be avoided through the proper implementation of currently available vaccination programs, with a substantial impact on reducing morbidity, mortality, and the economic burden on healthcare systems [4].

Scientific evidence demonstrates that vaccination is the most effective measure for preventing severe illness and mortality associated with influenza [5]. Similarly, vaccination programs have enabled the control, and even eradication, of numerous infectious diseases, such as smallpox, and significant control of measles, poliomyelitis, and rubella in various regions of the world [1]. Nevertheless, insufficient vaccination coverage can facilitate the resurgence of preventable disease outbreaks, related to low risk perception and prior negative experiences with vaccination [6].

Spain has a Common Vaccination and Immunization Schedule throughout life, approved by the Interterritorial Council of the National Health System (CISNS), which establishes specific age-based vaccination recommendations for the entire population [4,7]. This schedule includes essential vaccines against influenza, tetanus, diphtheria, pertussis, hepatitis B, and, more recently, COVID-19, with annual updates from the CISNS on objectives and guidelines. Nonetheless, misinformation and vaccine hesitancy have reduced coverage in multiple regions despite the availability of effective vaccines [5,8–11].

In 2015, the WHO, through the Strategic Advisory Group of Experts on Immunization (SAGE), indicated that vaccine hesitancy constitutes a complex and rapidly evolving global problem, and urged countries to identify contextual, individual, and vaccine-specific determinants influencing vaccination acceptance [12]. Vaccine hesitancy may also have been influenced by the rise of anti-vaccine movements in recent years, as well as the use of social media with the abundance of misinformation circulating on them [10,13,14].

Access to reliable information is crucial for an adequate perception of adult vaccination. In this regard, healthcare professionals play a central role in vaccine recommendations and the success of immunization campaigns [6,15], as well as an ethical responsibility to remain properly vaccinated to prevent transmission of diseases to vulnerable populations [5,16].

As future promoters of immunization, nursing students represent a key group in public health, although previous studies have identified misconceptions about the preventive role of vaccination [17–19]. Recent studies in Spain and Europe continue to document knowledge gaps and variable attitudes toward vaccination among nursing students, yet these have predominantly focused on pediatric or COVID-19 vaccination rather than adult schedules [20–22].

Although several national and international studies have examined vaccine knowledge and attitudes among nursing students [17–22], the majority have concentrated on childhood immunization, general hesitancy, or pandemic-specific vaccines. Few have specifically addressed knowledge of the adult vaccination schedule or explicitly examined the linkage between knowledge gaps, attitudes, and actual recommendation practices in a localized Spanish university context. Recent Spanish validation of a life-course immunization questionnaire among healthcare students [23] and cross-national comparisons of nursing students in Spain and Portugal [24] have begun to fill this gap, confirming persistent knowledge deficits even in final-year cohorts despite broadly positive attitudes. This regional gap limits the development of targeted curricular interventions tailored to the Spanish healthcare system.

The present study addresses these shortcomings by providing up-to-date, province-specific data from Alicante and directly linking knowledge deficiencies to supportive attitudes and recommendation behaviors among undergraduate nursing students.

This study is grounded in the 5C psychological antecedents of vaccine hesitancy model [25], which integrates five key determinants of vaccination-related behavior: confidence (trust in vaccine safety, efficacy, and providers), complacency (perceived low risk of disease), constraints (practical barriers), calculation (weighing risks and benefits), and collective responsibility (social norms and altruism). Knowledge of the adult vaccination schedule serves as a foundational precursor that influences confidence and calculation, while attitudes map directly onto confidence, complacency, and collective responsibility. This framework guided the selection of study variables and the design of the 19 questionnaire items (e.g., Likert-scale statements on perceived safety/efficacy assess confidence; items on disease risk perception and recommendation practices assess complacency and collective responsibility). Based on this model and prior evidence, the following specific hypotheses were tested:

H1: Knowledge of the Spanish adult vaccination schedule will increase significantly with advancing academic year (as a proxy for cumulative educational exposure).

H2: Greater knowledge will be positively associated with more favorable attitudes toward adult vaccination (higher confidence and lower complacency).

H3: More positive attitudes will be positively associated with stronger willingness to recommend all scheduled adult vaccines (greater collective responsibility).

The objective of this study is to explore the knowledge and attitudes of nursing students in the province of Alicante (Spain) regarding adult vaccination and its recommendation, in order to identify training needs and guide educational strategies that reinforce vaccine knowledge and predisposition to vaccination.

## 2. Methods and materials

### 2.1. Study design, setting, period, and sample size

An observational, descriptive, cross-sectional study was conducted among students from the three universities in the province of Alicante (Spain) that offer the Nursing degree: Universidad CEU Cardenal Herrera, Universidad Europea de Valencia, and Universidad de Alicante.

Inclusion criteria were active enrollment in any year of the Nursing degree program during the 2024/2025 academic year and voluntary consent to participate. There were no exclusion criteria other than failure to meet the inclusion criteria. A convenience sampling approach was used: all eligible students present in scheduled classroom sessions were invited to participate.

The study sample comprised all nursing students enrolled in any year of the degree program during the 2024/2025 academic year at any of the universities in the province of Alicante.

We calculated the minimum required sample size for this cross-sectional descriptive study using the formula for estimating proportions in a finite population, where N = 1189 (total population), Z = 1.96 (corresponding to a 95% confidence level), E = 0.05 (5% margin of error), and p = 0.5 (conservative estimate to maximize variance in the absence of prior proportion data). The total population was 1189 students, representing the university enrollment in the nursing degree for the 2024–2025 academic year, with 712 students from the Universidad de Alicante, 477 from the Universidad CEU Cardenal Herrera, and 72 from the Universidad Europea de Valencia. This yielded a minimum sample size of 292 participants.

## 2.2. Study tool (Questionnaire)

Data were collected using a 19-item ad hoc electronic questionnaire specifically developed for this study via Google Forms. The instrument was adapted from two previously validated and published tools in Spanish populations: the Vaccine Hesitancy Scale of the Spanish Society of Epidemiology [26] and the Vaccination Questionnaire for Health Sciences Students designed and validated by Fernández-Prada et al. [27]. Because both source instruments were already culturally and linguistically appropriate for Spanish university students, only minor wording adjustments were made to reference the current national adult vaccination schedule; these were verified through forward-back translation and consensus by two independent vaccinology experts.

A pilot study was conducted with 30 nursing students (excluded from the final analysis). This pilot assessed comprehension, user-friendliness, technical functionality, and time required for completion, while also confirming face and content validity through expert review. Construct validity was supported by known-groups discrimination. Internal consistency of the final instrument was evaluated using Cronbach's alpha, yielding acceptable values (α = 0.79 for the knowledge subscale and α = 0.85 for the attitudes/recommendation subscale). Discrimination validity was evidenced by the instrument's ability to differentiate between groups with varying levels of educational exposure: knowledge of the adult vaccination schedule and training scores differed significantly across academic years (p < 0.001), as shown in the Results section (Tables 2 and 3).

The final questionnaire consisted of 19 mandatory closed-ended items structured into three sections: (1) sociodemographic variables (6 items); (2) knowledge of the adult vaccination schedule and training received on vaccines (4 items); and (3) opinions and attitudes toward vaccination and its recommendation (9 items).

The survey was distributed between April 28 and May 24, 2025, through in-person visits to university classrooms. Researchers explained the study objectives, provided information on voluntary participation and data protection, and facilitated access via a QR code. The questionnaire was fully anonymous, with no collection of identifiable data, email addresses, or institutional affiliations. The header informed participants of the study's voluntary nature, ethical approval, and data protection in accordance with current regulations.

All questions were closed-ended and mandatory, resulting in zero incomplete responses. Participants were also informed of the study's main objective and that acceptance to complete the survey implied only its completion (S1 File).

## 2.3. Study variables

Sociodemographic variables: age (continuous), sex (male/female), marital status (single/ married or cohabiting), academic year (1st to 4th), and prior healthcare experience (yes/no, with years if applicable).

Knowledge variables: awareness of the existence of the Spanish adult vaccination schedule (yes/no/do not know); awareness of the specific vaccines included (yes/no, conditional). Training received on vaccines (5-point ordinal scale: nothing at all to a lot). Perception of information provided to the public (yes/no/do not know).

Attitude and recommendation variables: overall opinion on vaccines (4-point scale), perceived safety (5-point), perceived effectiveness (5-point), support for mandatory vaccination (compulsory/voluntary), willingness to recommend vaccines (no/ yes to a specific vaccine/ yes to all), and perceived resistance from other professionals (yes/no/do not know), with reasons for resistance (multiple response).

No composite numerical knowledge or attitude scores were calculated, as the items assess distinct constructs. Variables were analyzed as categorical or ordinal according to their original coding.

### 2.4. Ethical approval and informed consent

Positive evaluation was obtained from the Ethics Committee for Biomedical Experimentation of the Universidad CEU Cardenal Herrera (reference CEEI23/426). The study was conducted in accordance with the research principles established in the Declaration of Helsinki (October 2013), under the framework of good clinical practice guidelines (Order SCO256/2007, BOE13-II-2007) and Royal Decree 1090/2015 of December 24. Informed written consent was obtained from all participants. At the beginning of the electronic questionnaire (S1 File), a mandatory checkbox stated: "By ticking this box you consent to participation in the study. Furthermore, you declare that you are informed of your rights of access and information, rectification, deletion, oblivion, limitation of processing, data portability and objection, which you may exercise by writing to: jesus.sanchez@uchceu.es". Participants were also provided with a detailed information sheet (included in the questionnaire header) explaining the study objectives, voluntary nature of participation, anonymity, data-protection rights, and the possibility of withdrawing at any time without consequences. No minors were included in the study; all participants were undergraduate nursing students enrolled in higher-education institutions. The need for consent was not waived by the ethics committee. Extracted data were identified using a code, ensuring no inclusion of information that could identify participants. All collected data were stored on password-protected servers accessible only to the principal investigators and will be retained for five years after publication, after which they will be permanently and securely deleted, in accordance with the General Data Protection Regulation (EU) 2016/679 and Spanish Organic Law 3/2018 on the Protection of Personal Data. The authors have declared that no competing interests exist. No external funding was received for this study.

### 2.5. Statistical analysis

Data are expressed as mean ± standard deviation for continuous variables (age) and as frequencies and percentages for categorical and ordinal variables. The normality of age was assessed using the Kolmogorov-Smirnov test. Differences in age across academic years were evaluated with the non-parametric Kruskal-Wallis test, given the non-normal distribution. For categorical and ordinal variables, associations across academic years were examined using contingency tables and Pearson's $\chi^2$ test. All analyses were performed using IBM SPSS Statistics version 29.0. Statistical significance was set at $p < 0.05$. There were no missing data, as all questionnaire items were mandatory.

## 3. Results

### 3.1. Demographic data

A total of 562 nursing students responded to the survey, with first-year students being the most represented group and third-year students showing the lowest participation. All received questionnaires were valid. The participant population was predominantly female, single, and only 25.4% reported prior healthcare experience (Table 1). Statistical analysis indicated that all qualitative variables (sex, marital status, or healthcare experience) exhibited a homogeneous distribution across the different academic years of the degree program.

**Table 1. Characteristics of nursing students who participated in the study.**

|  | Total | Academic year | | | | p |
|  |  | 1st | 2nd | 3rd | 4th |  |
|---|---|---|---|---|---|---|
| **n (%)** | 562 (100) | 185 (32.9) | 151 (26.9) | 112 (19.9) | 114 (20.3) |  |
| **Age** | 23.0±6.3 | 21.3±6.7 | 22.8±5.4 | 23.8±5.6 | 25.5±6.7 | <0.001 |
| **Sex** |  |  |  |  |  |  |
| Male | 126 (22.4) | 42 (22.7) | 29 (19.2) | 32 (28.6) | 23 (20.2) | 0.299 |
| Female | 436 (77.6) | 143 (77.3) | 122 (80.8) | 80 (71.4) | 91 (79.8) |  |
| **Marital status** |  |  |  |  |  |  |
| Single | 453 (80.6) | 150 (81.1) | 128 (84.8) | 90 (80.4) | 85 (74.6) | 0.224 |
| Married/cohabiting couple | 109 (19.4) | 35 (18.9) | 23 (21.1) | 22 (19.6) | 29 (25.4) |  |
| **Healthcare experience** |  |  |  |  |  |  |
| No | 419 (74.6) | 150 (81.1) | 113 (74.8) | 76 (67.9) | 80 (70.2) | 0.047 |
| Yes | 143 (25.4) | 35 (18.9) | 38 (25.2) | 36 (32.1) | 34 (29.8) |  |

Quantitative variables are presented as mean±standard deviation. Qualitative variables are presented as number (%). p-value calculated using the Chi-square test for categorical variables and the Kruskal-Wallis test for the age variable.

## 3.2. Knowledge and training received on vaccines

Overall, 39.1% of the students were unaware of the existence of the adult vaccination schedule in Spain, and 8.9% denied its existence. This lack of awareness was higher in first-year students (55.7%) and lower in fourth-year students (27.2%; p<0.001), although 36% of final-year students expressed doubts or were unaware of it (Table 2).

**Table 2. Knowledge of the vaccination schedule and training received on vaccines.**

|  | Total | Academic year | | | | p |
|  |  | 1st | 2nd | 3rd | 4th |  |
|---|---|---|---|---|---|---|
| **Is there a vaccination schedule developed by the Spanish Ministry of Health for the adult population?** |  |  |  |  |  |  |
| Yes | 292 (52.0) | 70 (37.8) | 77 (51.0) | 72 (64.3) | 73 (64.0) | <0.001 |
| No | 50 (8.9) | 12 (6.5) | 20 (13.2) | 8 (7.1) | 10 (8.8) |  |
| Do not know | 220 (39.1) | 103 (55.7) | 54 (35.8) | 32(28.6) | 31 (27.2) |  |
| **If yes, are you aware of the vaccines included in the vaccination schedule for adults? (n=292)** |  |  |  |  |  |  |
| Yes | 142 (48.6) | 25 (35.7) | 37 (48.1) | 40 (55.6) | 40 (54.8) | 0.066 |
| No | 150 (51.4) | 45 (64.3) | 40 (51.9) | 32 (44.4) | 33 (45.2) |  |
| **How would you rate the training you have received so far in your degree programme on vaccines?** |  |  |  |  |  |  |
| Nothing at all | 38 (6.8) | 29 (15.7) | 9 (6.0) | 0 (0) | 0 (0) | <0.001 |
| Something | 119 (21.2) | 52 (28.1) | 20 (13.2) | 24 (21.4) | 23 (20.2) |  |
| Neither much nor little | 157 (27.9) | 59 (31.9) | 33 (21.9) | 30 (26.8) | 35 (30.7) |  |
| Quite a lot | 194 (34.5) | 33 (17.8) | 69 (45.7) | 44 (39.3) | 48 (42.1) |  |
| A lot | 54 (9.6) | 12 (6.5) | 20 (13.2) | 14 (12.5) | 8 (7.0) |  |
| **Do you believe that the general public receives sufficient information about vaccines before they are administered?** |  |  |  |  |  |  |
| Yes | 43 (7.7) | 16 (8.6) | 9 (6.0) | 12 (10.7) | 6 (5.3) | 0.044 |
| No | 485 (86.3) | 151 (81.6) | 132 (87.4) | 96 (85.7) | 106 (93.0) |  |
| Do not know | 34 (6.0) | 18 (9.7) | 10 (6.6) | 4 (3.6) | 2 (1.8) |  |

Data shown as a number (%). p-value calculated using the Chi-square test.

Among students who affirmed knowledge of the vaccination schedule, 51.4% did not identify the specific vaccines included, with a similar distribution of this information gap across the different academic years.

Only 44.1% of the students reported having received considerable or extensive training on vaccines during their degree program, with results suggesting that such training begins from the second year onward. Among fourth-year students, 50.9% indicated deficiencies in the vaccine training received. Furthermore, 86.3% of the total student body considered the information provided to the population prior to vaccine administration to be insufficient, a perception that intensified in higher years and reached 93% among final-year students.

In this regard, students were asked about the information sources they consulted regarding vaccines and those used by the general population for vaccine information (Fig 1). The most frequently cited source among students was their physician or nurse (54.1%), with a high percentage also reporting consultation with work colleagues (29.7%). Although they mentioned consulting institutional websites (27.8%) and accessing vaccine technical data sheets (23%), web search engines, forums, or blogs were the most commonly used internet sources among students (37.7%).

When asked about the general population, the majority of students believed that the public primarily relies on web search engines, forums, and blogs as sources of vaccine information (62.3%), with media and social networks being more frequently used than healthcare professionals themselves. Students perceived that the population rarely turns to institutional websites (7.3%) or official organizations specialized in vaccination to consult vaccine technical data sheets (8.4%). The complete response counts and percentages from the survey questions underlying Fig 1 are presented as supplementary data (S2 File).

### 3.3. Opinions on vaccines and attitudes toward vaccination recommendations

The general opinion on vaccines among the majority of nursing students was highly favorable (61.4%), with vaccines considered highly effective (62.5%) and quite safe (54.8%) (Table 3). Statistical analysis showed that students' general opinions on vaccines and vaccine safety were independent of their academic year.

First-year students exhibited a greater degree of skepticism regarding the efficacy of vaccines in disease prevention, although only 1.2% of the total surveyed students expressed an unfavorable attitude toward vaccines, and 0.5% considered them unsafe.

The majority of respondents (66.7%) believed that vaccines should be mandatory, and this attitude was not influenced by the students' academic level.

When asked about vaccine recommendations, the majority would recommend vaccination with all vaccines to the population (65.5%), and although this belief increased with training received during the degree program, no significant differences were observed based on the students' academic year.

Nevertheless, nearly half of the surveyed students (47.9%) reported having observed hesitancy among healthcare professionals in recommending vaccines. This perception was more frequent among higher-year students, reaching 70.2% among those in the final year.

The primary reasons for professional resistance to vaccine recommendations perceived by students were doubts about vaccine safety (40.9%) and efficacy (29.9%) (Fig 2), while concerns about the natural course of the disease (8%) or absence of belonging to a risk group (5.7%) were identified less frequently. The complete response counts and percentages from the survey questions underlying Fig 2 are presented as supplementary data (S2 File).

## 4. Discussion

Our findings reveal persistent deficiencies in knowledge of the adult vaccination schedule, with lower awareness in earlier academic years that only partially improves by the final year. These results are consistent with prior studies conducted in Spain, with no substantial improvement observed following the COVID-19 pandemic [20,21]. These findings align with available evidence from European studies. In Italy, nursing students exhibited significant knowledge gaps regarding

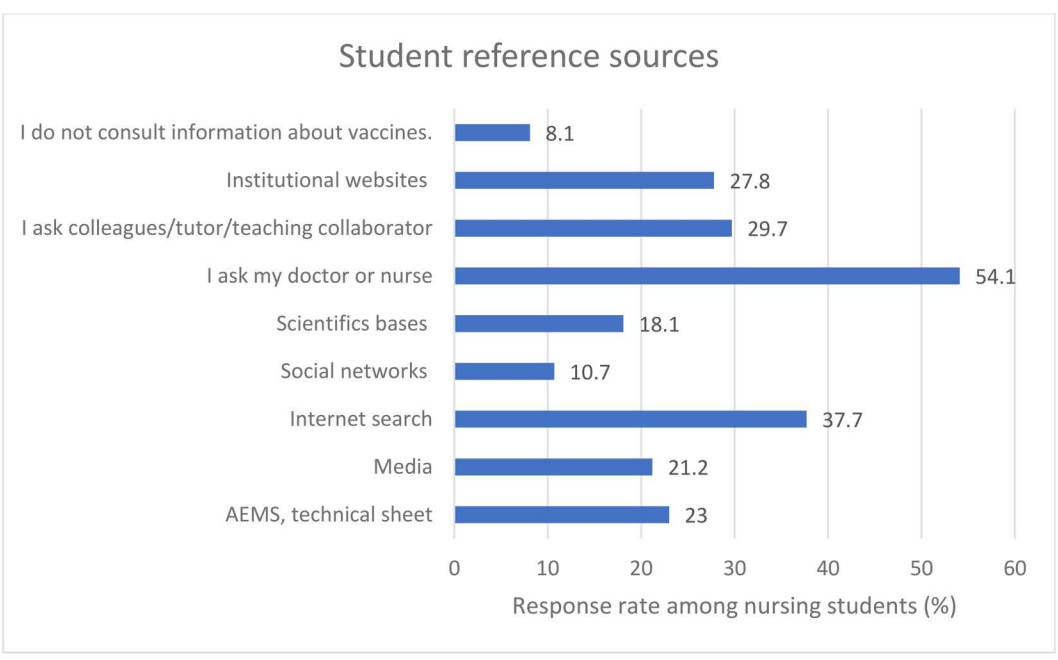

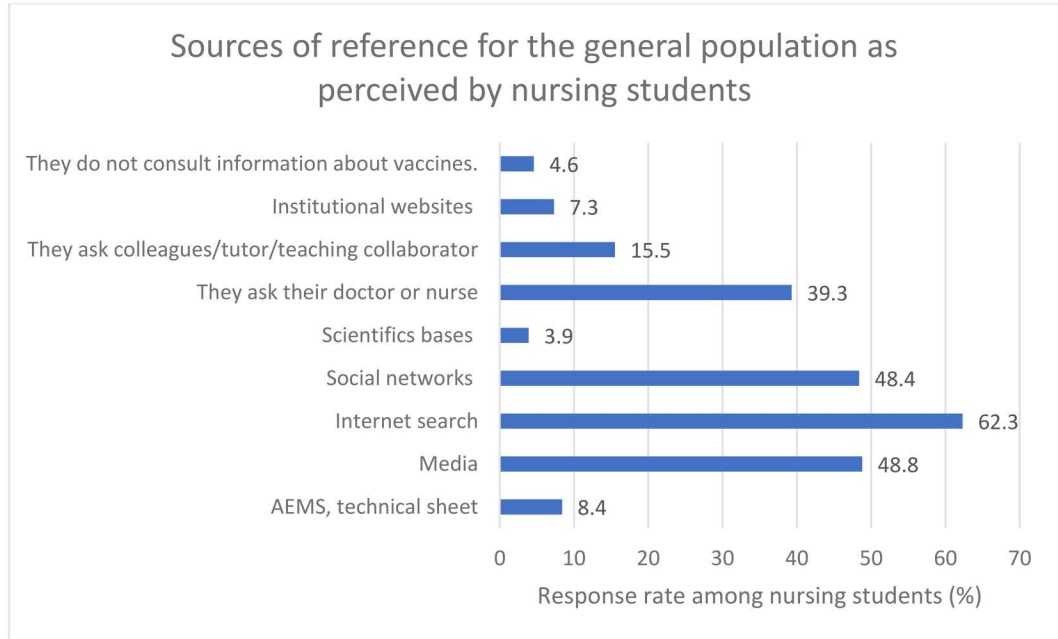

**Fig 1. Nursing students' self-reported sources of information on vaccinations and their perceptions of sources used by the general population.** Institutional websites: Ministry of Health, Autonomous Regions, International Agencies; Scientific bases: Pubmed, Web ofScience, Scopus, etc.; Social networks: YouTube, Twitter, Facebook, etc.; Internet search: websites, forums, blogs; Media: tv, press, magazines, etc.; AEMS: Spanish Agency for Medicines and Health Products.

**Table 3. General opinion on vaccines and their recommendation.**

| | Total | Academic year | | | | p |
| --- | --- | --- | --- | --- | --- | --- |
| | | 1st | 2nd | 3rd | 4th | |
| **Overall, your opinion on vaccines as a whole is:** | | | | | | |
| Not favourable | 7 (1.2) | 2 (1.1) | 3 (2.0) | 0 (0) | 2 (1.8) | 0.634 |
| Indifferent | 20 (3.6) | 10 (5.4) | 4 (2.6) | 2 (1.8) | 4 (3.5) | |
| Favourable | 190 (33.8) | 65 (35.1) | 47 (31.1) | 42 (37.5) | 36 (31.6) | |
| Very favourable | 345 (61.4) | 108 (58.4) | 97 (64.2) | 68 (60.2) | 72 (63.2) | |
| **Do you consider vaccines to be safe?** | | | | | | |
| Nothing at all | 3 (0.5) | 1 (0.5) | 1 (0.7) | 0 (0) | 1 (0.9) | 0.289 |
| Something | 21 (3.7) | 7 (3.8) | 8 (5.3) | 4 (3.6) | 2 (1.8) | |
| Neither much nor little | 45 (8.0) | 23 (12.4) | 10 (6.6) | 4 (3.6) | 8 (7.0) | |
| Quite a lot | 308 (54.8) | 98 (53.0) | 81 (53.6) | 70 (62.5) | 59 (51.8) | |
| A lot | 185 (32.9) | 56 (30.3) | 51 (33.8) | 34 (30.4) | 44 (38.6) | |
| **Do you consider vaccines to be effective in preventing disease?** | | | | | | |
| Nothing at all | 0 (0) | 0 (0) | 0 (0) | 0 (0) | 0 (0) | 0.034 |
| Something | 15 (2.7) | 9 (4.9) | 3 (2.0) | 0 (0) | 3 (2.6) | |
| Neither much nor little | 9 (1.6) | 5 (2.7) | 0 (0) | 4 (3.6) | 0 (0) | |
| Quite a lot | 187 (33.3) | 67 (36.2) | 45 (29.8) | 38 (33.9) | 37 (32.5) | |
| A lot | 351 (62.5) | 104 (56.2) | 103 (68.2) | 70 (62.5) | 74 (64.9) | |
| **In your opinion, the vaccines in the vaccination schedule should be** | | | | | | |
| Compulsory | 375 (66.7) | 127 (68.6) | 97 (64.2) | 74 (66.1) | 77 (67.5) | 0.853 |
| Voluntary | 187 (33.3) | 58 (31.4) | 54 (35.8) | 38 (33.9) | 37 (32.5) | |
| **As a future healthcare professional, would you recommend vaccination to your patients?** | | | | | | |
| No | 3 (0.5) | 2 (1.1) | 0 (0) | 0 (0) | 1 (0.9) | 0.200 |
| Yes, to a specific vaccine | 191 (34.0) | 70 (37.8) | 57 (37.7) | 34 (30.4) | 30 (26.3) | |
| Yes, to all vaccines | 368 (65.5) | 113 (61.1) | 94 (62.3) | 78 (69.6) | 83 (72.8) | |
| **Have you encountered resistance from other professionals when recommending vaccines?** | | | | | | |
| Yes | 269 (47.9) | 63 (34.1) | 68 (45.0) | 58 (51.8) | 80 (70.2) | <0.001 |
| No | 233 (41.5) | 87 (47.0) | 61 (40.0) | 54 (48.2) | 31 (27.2) | |
| Do not know | 60 (10.7) | 35 (18.9) | 22 (14.6) | 0 (0.0) | 3 (2.6) | |

Data shown as a number (%). p-value calculated using the Chi-square test.

vaccination [22], while in Turkey, pre-post educational interventions increased knowledge levels from 32% to 78% [28]. The results from our study on knowledge of the vaccination schedule remain concerning, with a consensus on the need for improved vaccinology training and updated curricula in nursing degrees in Spain, requirements that, as in our study, are demanded by the students themselves [20].

Participating students showed predominantly positive attitudes toward adult vaccination, viewing vaccines as both effective and safe. This favorable disposition was independent of academic year and formal training received. These results align with previous Spanish studies reporting high acceptance of vaccination among nursing students, who typically perceive vaccines as effective and safe [21].

Nevertheless, doubts persist in a minority of students, primarily related to vaccine efficacy or safety [29], as well as erroneous beliefs linking vaccination to chronic diseases, which highlight knowledge gaps [27]. In this regard, approximately 2% of students identify as anti-vaccine [30], a proportion comparable to our study.

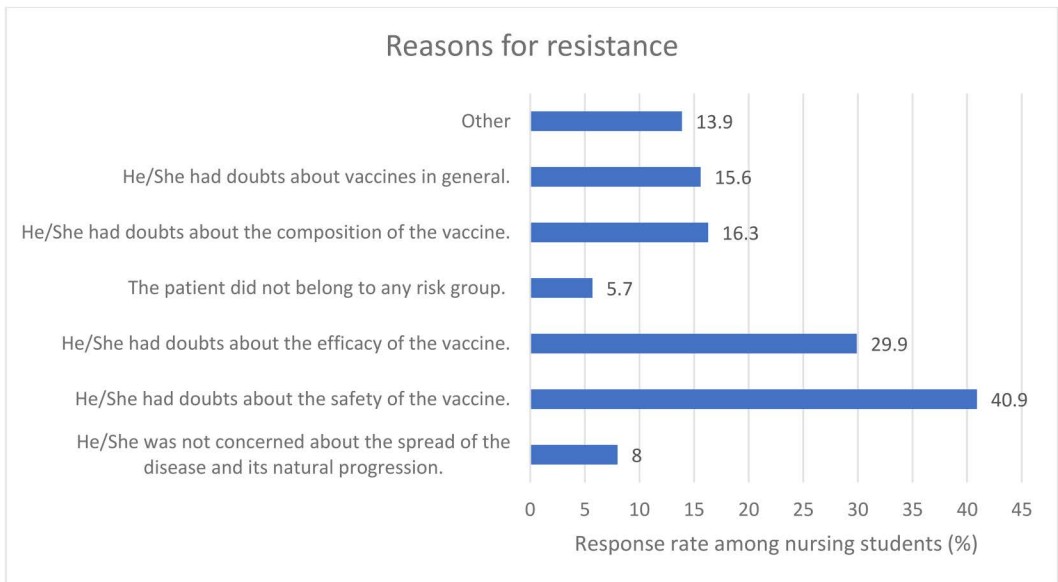

**Fig 2. Perception of reasons for resistance to vaccine recommendations by other health professionals detected by nursing students.** Data shown as a percentage of positive responses and calculated based on 269 students who reported encountering resistance from other professionals when recommending vaccines.

Educational interventions help shift these concerns toward more positive perspectives. Studies demonstrate that training eliminated beliefs about lack of safety, reinforcing the view of vaccines as the best prevention against vaccine-preventable diseases, as well as increasing willingness to vaccinate [31].

Health professionals and students show greater predisposition to abandon erroneous beliefs about vaccine harms, supported by scientific evidence confirming their benefits [32]. Accordingly, the nursing students in this study clearly acknowledged their future role as immunization promoters, demonstrating a strong willingness to recommend adult vaccines and widespread support for making them mandatory. These data highlight the importance of emphasizing the ethical aspects of vaccination, placing greater emphasis on the professional's informational responsibility over coercive approaches [33].

Nursing students reported observing resistance to vaccine recommendations among other healthcare professionals, with this perception becoming significantly more common as students progressed through the degree, especially in the final year. This may be due to nursing students gaining real-world practical healthcare experience as they progress through the degree, immersing themselves in a work environment and increasing contact with other healthcare providers.

In Spain, available data on healthcare professionals' hesitancy to recommend vaccines to the general population are limited and primarily focus on specific vaccines [5,11,34]. A key 2022 study on hesitancy toward the influenza vaccine, involving 11,108 healthcare professionals across Spain, used an online quantitative survey to identify contributing factors. This study showed that approximately 22% of healthcare professionals are unwilling to recommend vaccines from the vaccination schedule in general [33].

The primary reasons for this hesitancy include doubts about vaccine safety, efficacy, and necessity; low perception of disease risk; concerns about adverse effects; belief that the vaccine does not protect or even causes the disease; personal experiences with COVID-19 (such as illness or family losses); rejection of pharmaceutical practices or government decisions; and demographic factors such as gender, education, and income [33–35].

Prior vaccination and training in vaccinology were the most determinant factors for both personal vaccination and recommendation. In this regard, hesitancy to receive vaccines among healthcare professionals in Spain varies by vaccine

type and professional group. According to a recent study, approximately one third of healthcare professionals and health sciences students exhibit some degree of vaccine hesitancy in general, being higher among nursing staff, with the most rejected vaccines, in order, being: influenza, COVID-19, and human papillomavirus (HPV) [36]. The primary reasons are common in many cases: doubts about the necessity, safety, or effectiveness of vaccines; low perception of disease risk; post-pandemic vaccine fatigue; concerns about side effects or accelerated development (especially for COVID-19); and lack of effective awareness campaigns [36,37]. Improving training and communication could reduce doubts, as professionals' attitudes directly influence public acceptance. Opinions differ by profession, and higher rates among nurses could impact public recommendations due to their greater patient interaction, justifying the adoption of specific measures from their training to support their advisory role.

Only half of final-year students reported having received considerable or extensive training on vaccines during the degree program, and the vast majority consider that the general population does not receive sufficient information about vaccines before administration. This lack of information can lead both healthcare providers, as advisors and potential transmitters, and the general population to perceive vaccination erroneously [9]. In this regard, students indicated a preference for consulting physicians or nurses, as well as other work colleagues, and when using the internet, they prefer institutional websites. These results confirm prior studies suggesting that Spanish health professionals and students, such as nurses and physicians in training, obtain vaccine information primarily from academic and professional settings, including university curricula, scientific societies, and official health authorities, such as the Ministry of Health [20]. However, students believe that the population consults about vaccines mainly through unofficial search engines, social networks, and media outlets, primary vectors of misinformation [33]. This public misinformation or information from sources lacking unverified clinical rigor could result in lower vaccination rates. Thus, it is known that informal sources, such as family, friends, and social networks, are more frequent among groups that are hesitant or refuse vaccination, and more than a third of the general population trusts television or social networks, highlighting the potential risks of misinformation [6].

The present study makes a novel scientific contribution by providing the first province-specific evidence from Alicante on nursing students' knowledge of the complete Spanish adult vaccination schedule, while grounding the analysis in the 5C psychological antecedents of vaccine hesitancy model [25]. The dissociation between consistently positive attitudes toward adult vaccination and persistent knowledge deficits, evident even among fourth-year students, highlights a critical structural limitation in current nursing curricula. Although favorable opinions appear to emerge independently of formal training (potentially through societal exposure or early clinical contact), this decoupling reveals that attitudinal positivity alone is insufficient to equip future nurses as effective immunization promoters. The high rates of perceived resistance among practicing professionals reported by senior students further underscore how unaddressed knowledge gaps may perpetuate misinformation and undermine recommendation practices in real-world settings. These patterns align with recent European evidence indicating that nursing students frequently exhibit comparable shortfalls in vaccine literacy despite broadly positive dispositions [20–22].

Critically, the persistence of awareness gaps into the final year of training suggests that ad hoc or late-stage exposure to vaccinology content fails to produce the depth required for competent practice. Targeted educational interventions have demonstrated substantial efficacy in closing these gaps: quasi-experimental studies report large effect-size improvements in both vaccine knowledge and attitudes following structured modules, alongside significant reductions in hesitancy among nursing students [28]. Virtual programs incorporating case-based learning, role-playing, and communication skills training have yielded particularly robust gains in confidence for addressing patient concerns, with nursing cohorts showing the greatest reductions in hesitant attitudes [38].

These findings carry direct and actionable implications for the three nursing programs in Alicante. First, a mandatory, progressive vaccinology formation should be introduced from the first academic year, featuring interactive components such as simulation-based training on the adult immunization schedule, myth-debunking workshops, and evidence-based modules on communicating with vaccine-hesitant patients. Second, interdisciplinary standardization across the

participating universities would mitigate variations in teaching plans and ensure equitable knowledge acquisition. Finally, longitudinal evaluation of such interventions, extending into post-graduation practice, would generate robust evidence of their impact on immunization coverage and public health outcomes. These patterns are reinforced by a 2025 Spanish validation study of a life-course immunization questionnaire among healthcare students [23], which demonstrated high internal consistency and highlighted the need for explicit adult-schedule training, and by an international pilot on adult vaccine confidence among future health professionals that underscores the universality of the identified gaps [39]. Implementing these reforms would not only address the identified shortcomings but also strengthen the future workforce's capacity to counteract vaccine hesitancy and fulfill its ethical role in promoting adult immunization.

The study has some limitations that should be taken into account. One limitation of the study is that the results obtained when comparing across academic years may have been influenced by differences in the teaching plans among the three participating universities, as well as by the involvement of students who were not fully enrolled in the complete upper-level course at the time of completing the questionnaire.

A further limitation is that the ad hoc questionnaire did not yield a continuous or composite score for knowledge or attitudes toward adult vaccination. Consequently, results are presented as categorical frequencies and percentages rather than summarized numerical scores. This approach reflects the structure of the instrument but limits the possibility of more nuanced quantitative analyses (e.g., correlation or regression with total scores). Future studies should consider validated instruments that provide continuous knowledge and attitude scores when such metrics are required.

Another limitation is the external validity of the study, as the sample represents only nursing students from the Province of Alicante. However, this also constitutes a strength, since the high participation rate enhances internal validity and enables the derivation of valid conclusions with direct applicability for designing strategies related to the curriculum of the Nursing degree in the Province of Alicante.

## 5. Conclusions

This cross-sectional study among nursing students from the three universities offering the Nursing degree in the province of Alicante (Spain) identified a predominantly favorable attitude toward adult vaccination and its recommendation. However, significant knowledge gaps regarding the Spanish adult vaccination schedule persisted even in the final year of training. These findings are limited to the study population and setting and should not be extrapolated beyond nursing students in this geographic area.

The results highlight the need to strengthen and standardize vaccinology content within the curricula of these three specific institutions, particularly by introducing progressive training from the first academic year. Future longitudinal studies, ideally multicenter and extending into professional practice, are warranted to evaluate the effectiveness of such curricular modifications on knowledge retention, recommendation behaviors, and contribution to adult immunization coverage.

### Supporting information

**S1 File. Questionnaire.**
(DOCX)

**S2 File. Data related to Figures 1 and 2.**
(DOCX)

### Acknowledgments

We would like to thank the three Universities in the province of Alicante that offer Nursing degree (Universidad CEU Cardenal Herrera, Universidad Europea de Valencia and Universidad de Alicante) for allowing us access to classrooms to present the study and collect responses.

## Author contributions

**Conceptualization:** Noelia Rodríguez-Blanco, Cristina Salar-Andreu, Jesús Sánchez-Más, José Luis Duro-Torrijos.

**Data curation:** Noelia Rodríguez-Blanco, Nancy Vicente-Alcalde, Cristina Orts-Ruiz, Sergio Montero-Navarro.

**Formal analysis:** Jesús Sánchez-Más.

**Investigation:** Nancy Vicente-Alcalde, Cristina Orts-Ruiz, Sergio Montero-Navarro, Cristina Salar-Andreu.

**Methodology:** Nancy Vicente-Alcalde, Cristina Orts-Ruiz, Sergio Montero-Navarro, Cristina Salar-Andreu.

**Supervision:** Noelia Rodríguez-Blanco, Cristina Orts-Ruiz, Sergio Montero-Navarro, Cristina Salar-Andreu, Jesús Sánchez-Más, José Luis Duro-Torrijos.

**Validation:** Nancy Vicente-Alcalde, Sergio Montero-Navarro, Cristina Salar-Andreu, José Luis Duro-Torrijos.

**Writing – original draft:** Noelia Rodríguez-Blanco, Jesús Sánchez-Más, José Luis Duro-Torrijos.

**Writing – review & editing:** Noelia Rodríguez-Blanco, Jesús Sánchez-Más, José Luis Duro-Torrijos.

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
