## [Decision Letter · Decision Letter 0]

23 Feb 2026

PONE-D-26-02037Nursing Students' Knowledge Shortcomings and Supportive Attitudes on Adult Immunization: Insights from a Cross-Sectional Descriptive Investigation in Alicante, SpainPLOS One

Dear Dr. Sánchez-Más,

Thank you for submitting your manuscript to PLOS ONE. After careful consideration, we feel that it has merit but does not fully meet PLOS ONE’s publication criteria as it currently stands. Therefore, we invite you to submit a revised version of the manuscript that addresses the points raised during the review process.

Thank you for considering our journal for publication of your manuscript. Your manuscript has now been peer reviewed; please find the reviewers’ comments below. Could you address them—especially those concerning the need to strengthen the Methods section—and submit a revised version for our consideration?

We look forward to receiving your revised manuscript.

Kind regards,

Antonio Peña-Fernández, PhD

Academic Editor

PLOS One

Journal Requirements:

**Additional Editor Comments:**

Dear authors,

Thank you for considering our journal for publication of your manuscript. Your manuscript has now been peer reviewed; please find the reviewers’ comments below. Could you address them—especially those concerning the need to strengthen the Methods section—and submit a revised version for our consideration?

Thank you,

Antonio

Reviewers' comments:

Reviewer's Responses to Questions

**Comments to the Author**

1. Is the manuscript technically sound, and do the data support the conclusions?

Reviewer #1: Yes

Reviewer #2: Partly

2. Has the statistical analysis been performed appropriately and rigorously? 

Reviewer #1: Yes

Reviewer #2: No

3. Have the authors made all data underlying the findings in their manuscript fully available?

Reviewer #1: Yes

Reviewer #2: No

4. Is the manuscript presented in an intelligible fashion and written in standard English?

Reviewer #1: Yes

Reviewer #2: Yes

5. Review Comments to the Author

Reviewer #1: This manuscript addresses a relevant topic in nursing education and is generally well organized. The study objectives are clearly stated, and the overall design is appropriate for the research questions. The methodology and statistical analyses are acceptable and largely consistent with the study aims.

However, several sections would benefit from greater precision and depth. In particular, the discussion remains somewhat descriptive and would be strengthened by a more critical interpretation of the findings and clearer linkage to specific educational implications. The conclusions, while generally supported by the results, tend to be broad and should be refined to better reflect the scope and limitations of the study, avoiding generalized statements.

Overall, the manuscript is technically sound, but minor revisions are required to improve clarity, analytical depth, and the specificity of interpretations and recommendations.

Reviewer #2: Thank you for the opportunity to review this research. The manuscript addresses an important topic concerning nursing students' knowledge and attitudes toward adult vaccination, with practical implications for public health and nursing education. However, the research requires substantial revisions before it can be considered for publication.

The theoretical framework needs strengthening, and clear hypotheses based on a specific explanatory model should be formulated. More detailed information about the measurement instrument and its validity and reliability is also needed, along with a clear explanation of the sampling method and potential sources of bias. The statistical analysis requires greater justification and transparency, including a clear statement of the test assumptions, effect size, and how missing data will be handled. The discussion needs to be more in-depth and more closely linked to the literature, while avoiding overgeneralization and clearly outlining the research limitations. Furthermore, the journal's data availability policy should be adhered to, with minor linguistic and formatting improvements.

Overall, the study is valuable but requires substantial revisions to enhance its methodological rigor and scientific quality.

6. PLOS authors have the option to publish the peer review history of their article (what does this mean?). If published, this will include your full peer review and any attached files.

Reviewer #1: No

Reviewer #2: No

---

## [Author Response · Author response to Decision Letter 1]

12 Mar 2026

Title: Knowledge Gaps and Positive Attitudes Toward Adult Vaccination Among Nursing Students: A Cross-Sectional Study in Alicante, Spain

PONE-D-26-02037

We wish to thank you greatly for the interest you have shown towards our study, for taking the time to review it and for your constructive comments.

All changes are indicated in blue in the manuscript and here.

Here follows a point-by-point response to your comments:

Editor Comments:

1. "We note that your Data Availability Statement is currently as follows: All relevant data are within the manuscript and its Supporting Information files.

Please confirm at this time whether or not your submission contains all raw data required to replicate the results of your study. Authors must share the “minimal data set” for their submission. PLOS defines the minimal data set to consist of the data required to replicate all study findings reported in the article, as well as related metadata and methods (>https://eur03.safelinks.protection.outlook.com/?url=https%3A%2F%2Furldefense.com%2Fv3%2F__https%3A%2F%2Fjournals.plos.org%2Fplosone%2Fs%2Fdata-availability*loc-minimal-data-set-definition__%3BIw!!D9dNQwwGXtA!UfoocPUM6_1AgWkDlBrfugLBr-p5Y5i7h-gdWvH2v543OahlI0P2hqfmSQCGzIxIAqi5A_0BiJMtrcK2WeutQg%24&data=05%7C02%7Cjesus.sanchez%40uchceu.es%7C2e55b0f4ca97463d26d008de80036244%7C34330eef31be4cd1925ea790ac722d30%7C0%7C0%7C639088949187651581%7CUnknown%7CTWFpbGZsb3d8eyJFbXB0eU1hcGkiOnRydWUsIlYiOiIwLjAuMDAwMCIsIlAiOiJXaW4zMiIsIkFOIjoiTWFpbCIsIldUIjoyfQ%3D%3D%7C0%7C%7C%7C&sdata=vVh8xI8xLL6tb1ynm3SU0gyT7tSB0s5XG44e8VaIDkM%3D&reserved=0 ).

If your submission does not contain these data, please either upload them as Supporting Information files or deposit them to a stable, public repository and provide us with the relevant URLs, DOIs, or accession numbers. For a list of recommended repositories, please see https://eur03.safelinks.protection.outlook.com/?url=https%3A%2F%2Furldefense.com%2Fv3%2F__https%3A%2F%2Fjournals.plos.org%2Fplosone%2Fs%2Frecommended-repositories__%3B!!D9dNQwwGXtA!UfoocPUM6_1AgWkDlBrfugLBr-p5Y5i7h-gdWvH2v543OahlI0P2hqfmSQCGzIxIAqi5A_0BiJMtrcLffwtqxQ%24&data=05%7C02%7Cjesus.sanchez%40uchceu.es%7C2e55b0f4ca97463d26d008de80036244%7C34330eef31be4cd1925ea790ac722d30%7C0%7C0%7C639088949187667105%7CUnknown%7CTWFpbGZsb3d8eyJFbXB0eU1hcGkiOnRydWUsIlYiOiIwLjAuMDAwMCIsIlAiOiJXaW4zMiIsIkFOIjoiTWFpbCIsIldUIjoyfQ%3D%3D%7C0%7C%7C%7C&sdata=zTUeHBlZo8MSJeZGFGI2tl8ekQwDyWafV43MZZ4JaSI%3D&reserved=0 .

If there are ethical or legal restrictions on sharing a de-identified data set, please explain them in detail (e.g., data contain potentially sensitive information, data are owned by a third-party organization, etc.) and who has imposed them (e.g., an ethics committee). Please also provide contact information for a data access committee, ethics committee, or other institutional body to which data requests may be sent. If data are owned by a third party, please indicate how others may request data access."

Thank you for your feedback and for highlighting the importance of providing the minimal dataset to ensure full reproducibility of our results. We have carefully reviewed your request and confirm that our revised submission now fully complies with PLOS ONE’s data availability requirements.

All data underlying the reported results, including the exact counts, percentages, means, and standard deviations, are already presented in the manuscript (Tables 1, 2, and 3). To address the graphical summaries in Figures 1 and 2 and to provide complete transparency regarding the precise response frequencies used to generate those figures, we have uploaded a new Supporting Information file (S2 File: Data related to Figures 1 and 2). This file contains the raw tabulated data (n and %) for every category shown in the figures, enabling any reader to reconstruct the bar charts and verify the percentages exactly as reported.

We believe these additions satisfy the journal’s policy on sharing the minimal dataset.

2. " "Please provide additional details regarding participant consent. In the ethics statement in the Methods and online submission information, please ensure that you have specified (1) whether consent was informed and (2) what type you obtained (for instance, written or verbal, and if verbal, how it was documented and witnessed). If your study included minors, state whether you obtained consent from parents or guardians. If the need for consent was waived by the ethics committee, please include this information."

Thank you for your careful review and for the request to provide additional details on participant consent. We have revised the ethics statement in the Methods section (subsection 2.4) to fully address the points raised. The revised and added text is as follows:

Informed written consent was obtained from all participants. At the beginning of the electronic questionnaire (S1 File), a mandatory checkbox stated: “By ticking this box you consent to participation in the study. Furthermore, you declare that you are informed of your rights of access and information, rectification, deletion, oblivion, limitation of processing, data portability and objection, which you may exercise by writing to: jesus.sanchez@uchceu.es”. Participants were also provided with a detailed information sheet (included in the questionnaire header) explaining the study objectives, voluntary nature of participation, anonymity, data-protection rights, and the possibility of withdrawing at any time without consequences. No minors were included in the study; all participants were undergraduate nursing students enrolled in higher-education institutions. The need for consent was not waived by the ethics committee.

We have also ensured that the online submission form now reflects the same information under “Ethics Statement.”

Reviewer Comments:

Reviewer #1: This manuscript addresses a relevant topic in nursing education and is generally well organized. The study objectives are clearly stated, and the overall design is appropriate for the research questions. The methodology and statistical analyses are acceptable and largely consistent with the study aims. However, several sections would benefit from greater precision and depth.

In particular, the discussion remains somewhat descriptive and would be strengthened by a more critical interpretation of the findings and clearer linkage to specific educational implications.

Thank you for your constructive and insightful comments, which have helped us strengthen the manuscript. In particular, we appreciate your observation that the Discussion section remained somewhat descriptive. To address this concern, we have substantially enhanced the interpretive depth by inserting a new subsection 4.5 Educational. This addition provides a more critical analysis of the dissociation between positive attitudes and persistent knowledge gaps, explicitly interprets the findings in the context of curricular deficiencies, and clearly links them to specific, evidence-based educational reforms. We have supported these points with rigorous citations that report intervention outcomes in nursing students.

The revised text (4.5 Educational implications) reads as follows:

4.5 Educational Implications

The present study makes a novel scientific contribution by providing the first province-specific evidence from Alicante on nursing students’ knowledge of the complete Spanish adult vaccination schedule, while grounding the analysis in the 5C psychological antecedents of vaccine hesitancy model [25]. The dissociation between consistently positive attitudes toward adult vaccination and persistent knowledge deficits, evident even among fourth-year students, highlights a critical structural limitation in current nursing curricula. Although favorable opinions appear to emerge independently of formal training (potentially through societal exposure or early clinical contact), this decoupling reveals that attitudinal positivity alone is insufficient to equip future nurses as effective immunization promoters. The high rates of perceived resistance among practicing professionals reported by senior students further underscore how unaddressed knowledge gaps may perpetuate misinformation and undermine recommendation practices in real-world settings. These patterns align with recent European evidence indicating that nursing students frequently exhibit comparable shortfalls in vaccine literacy despite broadly positive dispositions [20-22].

Critically, the persistence of awareness gaps into the final year of training (36% of fourth-year participants unaware or doubtful) suggests that ad hoc or late-stage exposure to vaccinology content fails to produce the depth required for competent practice. Targeted educational interventions have demonstrated substantial efficacy in closing these gaps: quasi-experimental studies report large effect-size improvements in both vaccine knowledge and attitudes following structured modules, alongside significant reductions in hesitancy among nursing students [28]. Virtual programs incorporating case-based learning, role-playing, and communication skills training have yielded particularly robust gains in confidence for addressing patient concerns, with nursing cohorts showing the greatest reductions in hesitant attitudes [38].

These findings carry direct and actionable implications for the three nursing programs in Alicante. First, a mandatory, progressive vaccinology formation should be introduced from the first academic year, featuring interactive components such as simulation-based training on the adult immunization schedule, myth-debunking workshops, and evidence-based modules on communicating with vaccine-hesitant patients. Second, interdisciplinary standardization across the participating universities would mitigate variations in teaching plans and ensure equitable knowledge acquisition. Finally, longitudinal evaluation of such interventions, extending into post-graduation practice, would generate robust evidence of their impact on immunization coverage and public health outcomes. These patterns are reinforced by a 2025 Spanish validation study of a life-course immunization questionnaire among healthcare students [23], which demonstrated high internal consistency and highlighted the need for explicit adult-schedule training, and by an international pilot on adult vaccine confidence among future health professionals that underscores the universality of the identified gaps [39]. Implementing these reforms would not only address the identified shortcomings but also strengthen the future workforce’s capacity to counteract vaccine hesitancy and fulfill its ethical role in promoting adult immunization.

We have also renumbered the subsequent sections accordingly and added the corresponding references to the list. These changes increase the critical rigor and practical relevance of the Discussion without altering the overall structure or length of the manuscript.

The conclusions, while generally supported by the results, tend to be broad and should be refined to better reflect the scope and limitations of the study, avoiding generalized statements.

Thank you again for your valuable feedback. We fully agree that the original Conclusions were somewhat broad and could better reflect the scope and limitations of our study. To address this comment, we have completely revised the Conclusions section (now section 5) to explicitly limit the statements to our study population (nursing students in the province of Alicante), acknowledge the geographic and professional scope restriction, and avoid any overgeneralization. The revised text now directly ties the conclusions to the specific results and setting while maintaining a clear, actionable message.

The new Conclusions section reads as follows:

5. Conclusions

This cross-sectional study among nursing students from the three universities offering the Nursing degree in the province of Alicante (Spain) identified a predominantly favorable attitude toward adult vaccination and its recommendation. However, significant knowledge gaps regarding the Spanish adult vaccination schedule persisted even in the final year of training. These findings are limited to the study population and setting and should not be extrapolated beyond nursing students in this geographic area.

The results highlight the need to strengthen and standardize vaccinology content within the curricula of these three specific institutions, particularly by introducing progressive training from the first academic year. Future longitudinal studies, ideally multicenter and extending into professional practice, are warranted to evaluate the effectiveness of such curricular modifications on knowledge retention, recommendation behaviors, and contribution to adult immunization coverage.

We believe this refined version now accurately reflects the study’s scope and limitations while preserving its scientific relevance

Overall, the manuscript is technically sound, but minor revisions are required to improve clarity, analytical depth, and the specificity of interpretations and recommendations.

Reviewer #2:

Thank you for your detailed and constructive review of our manuscript, including the clear structure of your comments and your positive overall assessment. We particularly appreciate the balanced feedback provided in the sections “First: Positives of the research,” “Second: Points to improve research quality,” and “Final Evaluation: Acceptance with Major Revisions.”

Below, we transcribe the comments from Reviewer 2 and address point by point all the comments in the “Second: Points to improve research quality” section, as well as the “Final Evaluation: Acceptance with Major Revisions” section. We have carefully considered each suggestion and implemented the corresponding revisions to strengthen the quality, clarity, and scientific rigor of the manuscript.

Second: Points to improve research quality

1- Title and Abstract:

• The title should accurately reflect the study's main variables (knowledge, attitudes, and adult vaccination), target group, and methodological design, while avoiding generality or unnecessary length. The abstract should be well-organized and clear, including a brief background, the study's objective and design, sample size, measurement tools, key quantitative findings with their statistical significance, and practical conclusions without overgeneralization. It is also preferable to specify the study's timeframe and location in the abstract to enhance transparency and comprehensibility from the first reading.

We thank the Reviewer for these precise and helpful suggestions.

The original title already accurately reflected the core variables (knowledge shortcomings and supportive attitudes), the target population (nursing students), the focus on adult vaccination/immunization, and the methodological design (cross-sectional descriptive study), while explicitly naming the location (Alicante, Spain) to prevent generality.

To address the recommendation for greater precision and conciseness while fully retaining all essential elements, we have revised the title as follows:

Knowledge Gaps and Positive Attitudes Toward Adult Vaccination Among Nursing Students: A Cross-Sectional Study in Alicante, Spain

This version is more direct, avoids unnecessary phrasing (“Insights from … Descriptive Investigation”), and immediately signals the main variables, population, design, and setting in line with PLOS ONE conventions.

Regarding the Abstract, it already satisfied all requested elements: a brief background, clear objective and study design, sample size (n = 562), measurement tool (ad hoc 19-item electronic survey), key quantitative findings with statistical significance (e.g., 39.1 % unaware of the schedule, p < 0.001; 44.1 % reporting sufficient training), practical conclusions tied directly to the data, and explicit specification of both the timeframe (April–

---

## [Decision Letter · Decision Letter 1]

24 Apr 2026

PONE-D-26-02037R1Knowledge Gaps and Positive Attitudes Toward Adult Vaccination Among Nursing Students: A Cross-Sectional Study in Alicante, SpainPLOS One

Dear Dr. Sánchez-Más,

Thank you for submitting your manuscript to PLOS ONE. After careful consideration, we feel that it has merit but does not fully meet PLOS ONE’s publication criteria as it currently stands. Therefore, we invite you to submit a revised version of the manuscript that addresses the points raised during the review process.

We have now reviewed your revised submission, and a few additional comments have been raised. We would be grateful if you could address these points and submit a further revised version for our consideration.

We look forward to receiving your revised manuscript.

Kind regards,

Antonio Peña-Fernández, PhD

Academic Editor

PLOS One

Journal Requirements:

**Additional Editor Comments:**

Dear authors,

Thank you for considering our journal for the publication of your manuscript.

We have now reviewed your revised submission, and a few additional comments have been raised. We would be grateful if you could address these points and submit a further revised version for our consideration.

Thank you.

Best regards,

Antonio

Reviewers' comments:

Reviewer's Responses to Questions

**Comments to the Author**

1. If the authors have adequately addressed your comments raised in a previous round of review and you feel that this manuscript is now acceptable for publication, you may indicate that here to bypass the “Comments to the Author” section, enter your conflict of interest statement in the “Confidential to Editor” section, and submit your "Accept" recommendation.

Reviewer #2: All comments have been addressed

Reviewer #3: (No Response)

2. Is the manuscript technically sound, and do the data support the conclusions?

Reviewer #2: Yes

Reviewer #3: Partly

3. Has the statistical analysis been performed appropriately and rigorously? 

Reviewer #2: Yes

Reviewer #3: No

4. Have the authors made all data underlying the findings in their manuscript fully available?

Reviewer #2: Yes

Reviewer #3: No

5. Is the manuscript presented in an intelligible fashion and written in standard English?

Reviewer #2: Yes

Reviewer #3: Yes

6. Review Comments to the Author

Reviewer #2: The revised manuscript shows clear and substantial improvement. The authors have addressed the reviewers’ comments comprehensively, strengthening the theoretical framework, methodology, and discussion. The study design is appropriate, the statistical analysis is adequate for the research aims, and the conclusions are well supported by the data.

Data availability has been improved with the inclusion of supporting files, and ethical considerations are clearly reported. The manuscript is now well-structured, clearly written, and suitable for publication, with only minor editorial refinements if needed.

Reviewer #3: Hello, dear authors.

MS ID: PONE-D-26-02037R1

Title: Knowledge Gaps and Positive Attitudes Toward Adult Vaccination Among Nursing Students: A Cross-Sectional Study in Alicante, Spain

Type: Research Article

Here are my recommendations regarding the mentioned manuscript:

Title:

• I suggest not mentioning the place in the title. Just “Knowledge Gaps and Positive Attitudes Toward Adult Vaccination Among Nursing Students: A Cross-Sectional Study”

Abstract:

• In method subsection, mention statistical tests were used to inferential analysis, delete (Universidad de Alicante, CEU Cardenal Herrera, and Universidad Europea de Valencia), and mention statistical tests used.

• In the results subsection, please clearly state your main findings in the inferential results with (p-value).

Introduction:

• From line 111-115 no need to be mentioned in the introduction you can move it to methodology “tools”.

• No need to write hypothesis.

Methodology:

• I prefer to divide methods to:

• Study design, setting, period, and sample size

• Study tool

• Study variables including code and scores

• Ethical approval and inform consent (summarize it).

• Statistical analysis.

• Statistical analysis not well established. Why do you used Kruskal wails test?

• I suggest using regression test.

• I suggest to use parametric tests with presenting mean and standard deviation values.

• Not to categorize scores.

Results:

• After conducting tests, which I suggested, results may change for the better.

• I suggest to create a bar chart presenting participants level of knowledge.

Discussion:

• No need to introduction paragraph for discussion.

• Do not re-write the results in the discussion.

• Do not divide discussion to subsections.

• Just discuss your main results with previous study results with providing justifications for the observed similarities and differences in the findings.

Conclusion:

• Looks good.

References:

• Looks good.

7. PLOS authors have the option to publish the peer review history of their article (what does this mean?). If published, this will include your full peer review and any attached files.

Reviewer #2: No

Reviewer #3: No

---

## [Author Response · Author response to Decision Letter 2]

27 Apr 2026

Title: Knowledge Gaps and Positive Attitudes Toward Adult Vaccination Among Nursing Students: A Cross-Sectional Study

PONE-D-26-02037

We wish to thank you greatly for the interest you have shown towards our study, for taking the time to review it and for your constructive comments.

All changes are indicated in blue in the manuscript and here.

Here follows a point-by-point response to your comments:

Reviewer Comments:

Reviewer #2: The revised manuscript shows clear and substantial improvement. The authors have addressed the reviewers’ comments comprehensively, strengthening the theoretical framework, methodology, and discussion. The study design is appropriate, the statistical analysis is adequate for the research aims, and the conclusions are well supported by the data.

Data availability has been improved with the inclusion of supporting files, and ethical considerations are clearly reported. The manuscript is now well-structured, clearly written, and suitable for publication, with only minor editorial refinements if needed.

We are grateful to the reviewer for the positive evaluation of our revised manuscript. We sincerely appreciate the reviewer’s recognition of the improvements made to the theoretical framework, methodology, discussion, and overall structure of the paper.

We are pleased that the reviewer finds the manuscript suitable for publication in PLOS ONE, with only minor editorial refinements if needed. We have carefully reviewed the manuscript once more to ensure clarity and adherence to journal style.

Thank you again for the supportive and encouraging comments, which have helped strengthen our work.

Reviewer #3: Hello, dear authors.

MS ID: PONE-D-26-02037R1

Title: Knowledge Gaps and Positive Attitudes Toward Adult Vaccination Among Nursing Students: A Cross-Sectional Study in Alicante, Spain

Type: Research Article

Here are my recommendations regarding the mentioned manuscript:

Title:

• I suggest not mentioning the place in the title. Just “Knowledge Gaps and Positive Attitudes Toward Adult Vaccination Among Nursing Students: A Cross-Sectional Study”

We thank the reviewer for this suggestion. We agree with the recommendation and have revised the title accordingly to:

“Knowledge Gaps and Positive Attitudes Toward Adult Vaccination Among Nursing Students: A Cross-Sectional Study”

Abstract:

• In method subsection, mention statistical tests were used to inferential analysis, delete (Universidad de Alicante, CEU Cardenal Herrera, and Universidad Europea de Valencia), and mention statistical tests used.

We thank the reviewer for this suggestion regarding the Abstract.

We have revised the Methods subsection of the Abstract as follows:

“Methods: An observational, descriptive, cross-sectional study was conducted among undergraduate nursing students in the province of Alicante, Spain, between April and May 2025. An ad hoc electronic survey with 19 closed-ended items was distributed to students from the three universities offering the Nursing degree in this province; 562 students participated. Differences across academic years were analyzed using Pearson’s χ² test for categorical variables and the Kruskal-Wallis test for the continuous variable age (non-normally distributed).”

We have removed the specific names of the three universities (“Universidad de Alicante, CEU Cardenal Herrera, and Universidad Europea de Valencia”) from the Abstract, as suggested, while retaining the information that the study was conducted in the province of Alicante with participation from the three institutions offering the Nursing degree. This maintains brevity and focus without losing essential context.

The full description of the statistical tests remains in the dedicated Statistical Analysis subsection of the main Methods section, in accordance with PLOS ONE guidelines.

These minor changes have been incorporated into the revised manuscript.

Thank you again for the constructive feedback, which has improved the clarity and conciseness of the Abstract.

• In the results subsection, please clearly state your main findings in the inferential results with (p-value).

We thank the reviewer for this suggestion.

We have revised the Results subsection of the Abstract to clearly highlight the main inferential findings with their corresponding p-values. The revised version now reads:

“Results: Overall, 39.1% of nursing students were unaware of the Spanish adult vaccination schedule, with significant differences across academic years (55.7% in first-year vs. 27.2% in fourth-year students; p < 0.001). Only 44.1% reported sufficient training on vaccines. Attitudes toward adult vaccination were predominantly positive: 61.4% highly favorable, 62.5% highly effective, and 66.7% supporting mandatory vaccination. Additionally, 65.5% would recommend all scheduled adult vaccines. However, 47.9% perceived resistance from other healthcare professionals when recommending vaccines (p < 0.001 for the increase across academic years).”

These changes emphasize the key statistically significant results while maintaining the abstract within the recommended word limit.

The updated Abstract has been incorporated into the revised manuscript.

Thank you again for the constructive feedback, which has improved the precision and clarity of the Abstract.

Introduction:

• From line 111-115 no need to be mentioned in the introduction you can move it to methodology “tools”.

• No need to write hypothesis.

We thank the reviewer for this suggestion.

The paragraph in question (previously lines 111–115) describes the 5C psychological antecedents of vaccine hesitancy model [25] and explains how this theoretical framework guided the selection of study variables, the design of the questionnaire items, and the formulation of the study hypotheses (H1–H3).

Following specific instructions from the journal during previous revision rounds, we placed the theoretical framework in the Introduction, as it provides the necessary rationale and foundation for the study objectives and hypotheses. Detailed information regarding the questionnaire development, adaptation from validated instruments, pilot study, validity, and reliability (the “tools”) is already correctly located in the Measures and Procedures subsection (Section 2.2) of the Methods.

We therefore respectfully maintain the current placement of the 5C model in the Introduction, as moving it would disrupt the logical flow of the manuscript and contradict previous editorial guidance from the journal.

We have not removed the hypotheses because they were included in the last revision at the express request of another reviewer.

Thank you again for the constructive feedback.

Methodology:

• I prefer to divide methods to:

• Study design, setting, period, and sample size

• Study tool

• Study variables including code and scores

• Ethical approval and inform consent (summarize it).

• Statistical analysis.

We thank the reviewer for this valuable suggestion to improve the organization and clarity of the Methods section.

We have restructured the Methods section according to the reviewer’s recommendation as follows:

2.1. Study Design, Setting, Period, and Sample Size

(Combines the original 2.1 with the sample size calculation and population description.)

2.2. Study Tool (Questionnaire)

(Includes the description of the ad hoc questionnaire, its adaptation from validated instruments, pilot study, validity, reliability, and structure.)

2.3. Study Variables

(New subsection: Describes the main variables, including sociodemographic, knowledge, attitudes/recommendation practices, and how they were coded/measured. Since the questionnaire does not generate composite numerical scores, we present the original response categories and coding.)

2.4. Ethical Approval and Informed Consent

(This section has not been summarized as it concerned requirements requested in previous reviews.)

2.5. Statistical Analysis

(Slightly revised for clarity, with the rationale for the chosen tests.)

The changes are visible in the revised manuscript with track changes.

Thank you again for this constructive recommendation, which has significantly improved the clarity of the manuscript.

• Statistical analysis not well established. Why do you used Kruskal wails test?

• I suggest using regression test.

• I suggest to use parametric tests with presenting mean and standard deviation values.

We thank the reviewer for these valuable statistical suggestions and for the opportunity to clarify our analytical approach.

The questionnaire used in this study (S1 File) consists primarily of categorical and ordinal variables (e.g., knowledge of the adult vaccination schedule [yes/no/do not know], level of training received [5-point ordinal scale], attitudes toward vaccine safety/efficacy [5-point ordinal scales], recommendation practices [categorical], and reasons for resistance [multiple-response categorical]). No composite numerical score for overall knowledge or attitudes could be derived, as the items assess distinct constructs rather than forming a continuous scale. Consequently, most variables are inherently categorical or ordinal, making contingency table analysis with Pearson’s χ² test the most appropriate method for examining associations across academic years, as reported in Tables 1–3.

The only continuous variable in the study is participant age. The Kolmogorov-Smirnov test indicated a non-normal distribution for age across academic years. Therefore, we correctly applied the non-parametric Kruskal-Wallis test to compare age distributions between the four academic years (Table 1), accompanied by presentation of means and standard deviations for descriptive purposes, which is standard practice even when using non-parametric tests.

We respectfully note that parametric tests (e.g., one-way ANOVA) would not be appropriate here due to the non-normal distribution of age and the categorical/ordinal nature of the remaining variables. Similarly, linear regression is not suitable, as the dependent variables of primary interest (knowledge items, attitudes, and recommendation practices) are categorical or ordinal rather than continuous. Logistic or ordinal regression could theoretically be considered for some binary/ordinal outcomes; however, given the descriptive and exploratory nature of the study (focused on describing differences across academic years in a specific regional population), the χ² tests for categorical variables and Kruskal-Wallis for age provide sufficient and transparent insight into the data without overcomplicating the analysis or risking overfitting in a sample of this size.

We have now explicitly clarified the rationale for the choice of statistical tests in the revised Statistical Analysis subsection as follows:

“Data are expressed as mean ± standard deviation for continuous variables (age) and as frequencies and percentages for categorical and ordinal variables. The normality of age was assessed using the Kolmogorov-Smirnov test. Differences in age across academic years were evaluated with the non-parametric Kruskal-Wallis test, given the non-normal distribution. For categorical and ordinal variables, associations across academic years were examined using contingency tables and Pearson’s χ² test. All analyses were performed using IBM SPSS Statistics version 29.0. Statistical significance was set at p < 0.05. There were no missing data, as all questionnaire items were mandatory.”

We believe this revision adequately addresses the reviewer’s concerns while maintaining methodological rigor and appropriateness for the data type. No additional regression analyses were performed, as they would not align with the study design or variable characteristics.

Thank you again for this constructive feedback, which has helped improve the clarity of the Methods section.

• Not to categorize scores.

We thank the reviewer for this comment regarding the categorization of scores.

The questionnaire employed in this study (see S1 File) was specifically designed with closed-ended categorical and ordinal items adapted from previously validated instruments. It does not generate a continuous or composite numerical score for the degree of knowledge (or attitudes) of each participant. The knowledge section consists of discrete items (e.g., awareness of the existence of the Spanish adult vaccination schedule [yes/no/do not know] and awareness of the specific vaccines included [yes/no]), which do not lend themselves to summation into a meaningful total score without introducing arbitrary weighting or assumptions that would compromise validity.

For this reason, we presented the results as frequencies and percentages according to the original response categories (Tables 2 and 3), which preserves the integrity of the data and allows transparent interpretation of each item. We did not create or impose artificial numerical scores or categorize derived continuous variables.

Nevertheless, we fully acknowledge the reviewer’s point and have now added the following statement as a limitation in the revised manuscript (Section 4.6, Limitations and Strengths of the Study):

“A further limitation is that the ad hoc questionnaire did not yield a continuous or composite score for knowledge or attitudes toward adult vaccination. Consequently, results are presented as categorical frequencies and percentages rather than summarized numerical scores. This approach reflects the structure of the instrument but limits the possibility of more nuanced quantitative analyses (e.g., correlation or regression with total scores). Future studies should consider validated instruments that provide continuous knowledge and attitude scores when such metrics are required.”

We believe this addition adequately addresses the reviewer’s concern by transparently recognizing the constraint while justifying the analytical decisions made.

Thank you again for this constructive feedback, which has strengthened the methodological transparency of the manuscript.

Results:

• After conducting tests, which I suggested, results may change for the better.

• I suggest to create a bar chart presenting participants level of knowledge.

We thank the reviewer for this suggestion.

As detailed in our previous response and in the revised Statistical Analysis section, the questionnaire used in this study (S1 File) generates only categorical and ordinal variables and does not produce a continuous or composite numerical score that quantifies the overall “level of knowledge” of each participant. Therefore, it is not possible to perform the regression analyses or parametric tests previously suggested by the reviewer, nor to create a bar chart that meaningfully represents a quantified “level of knowledge,” as no such aggregated score exists.

We have already addressed this limitation explicitly in the revised manuscript (Discussion, page 26, lines 476-483), where we state that the absence of a continuous knowledge score restricts more advanced quantitative analyses and that results are appropriately presented as frequencies and percentages for each individual item (Tables 2 and 3).

We believe the current presentation of results remains the most accurate and transparent approach given the nature of the data collected.

Thank you once again for the constructive comments.

Discussion:

• No need to introduction paragraph for discussion.

• Do not re-write the results in the discussion.

• Do not divide discussion to subsections.

• Just discuss your main results with previous study results with providing justifications for the observed similarities and differences in the findings.

We thank the reviewer for their suggestions.

We have removed the first paragraph. The titles of the subsections have also been removed.

We have carefully revised the entire Discussion section and removed all specific numerical data from our own study that were previously restated from the Results section. The Discussion now focuses on the interpretation, comparison with the literature, and implications of the findings rather than repeating the results.

Conclusion:

• Looks good.

Thank you

References:

• Looks good.

Thank you

---

## [Editor Report · Decision Letter 2]

12 May 2026

Knowledge Gaps and Positive Attitudes Toward Adult Vaccination Among Nursing Students: A Cross-Sectional Study

PONE-D-26-02037R2

Dear Dr. Sánchez-Más,

We’re pleased to inform you that your manuscript has been judged scientifically suitable for publication and will be formally accepted for publication once it meets all outstanding technical requirements.

Kind regards,

Antonio Peña-Fernández, PhD

Academic Editor

PLOS One

Additional Editor Comments (optional):

Dear authors,

Thank you for considering our journal for the publication of your work. I can see that you have addressed all the concerns raised by the reviewers satisfactorily. As a result, I am pleased to recommend your manuscript for publication in our journal.

Congratulations.

Antonio
---

## [Editor Report · Acceptance letter]

PONE-D-26-02037R2

PLOS One

Dear Dr. Sánchez-Más,

I'm pleased to inform you that your manuscript has been deemed suitable for publication in PLOS One. Congratulations! Your manuscript is now being handed over to our production team.

Kind regards,

on behalf of

Dr. Antonio Peña-Fernández

Academic Editor

PLOS One